# Leadership Effectiveness in Healthcare Settings: A Systematic Review and Meta-Analysis of Cross-Sectional and Before–After Studies

**DOI:** 10.3390/ijerph191710995

**Published:** 2022-09-02

**Authors:** Vincenzo Restivo, Giuseppa Minutolo, Alberto Battaglini, Alberto Carli, Michele Capraro, Maddalena Gaeta, Anna Odone, Cecilia Trucchi, Carlo Favaretti, Francesco Vitale, Alessandra Casuccio

**Affiliations:** 1Department of Health Promotion, Maternal and Infant Care, Internal Medicine and Medical Specialties (PROMISE) “G. D’Alessandro”, University of Palermo, Via del Vespro 133, 90127 Palermo, Italy; 2Vaccines and Clinical Trials Unit, Department of Health Sciences, University of Genova, Via Antonio Pastore 1, 16132 Genova, Italy; 3Santa Chiara Hospital, Largo Medaglie d’oro 9, 38122 Trento, Italy; 4School of Public Health, Vita-Salute San Raffaele University, Via Olgettina 58, 20132 Milan, Italy; 5Department of Public Health, Experimental and Forensic Medicine, University of Pavia, Via Forlanini 2, 27100 Pavia, Italy; 6Planning, Epidemiology and Prevention Unit, Liguria Health Authority (A.Li.Sa.), IRCCS San Martino Hospital, Largo R. Benzi 10, 16132 Genoa, Italy; 7Centre on Leadership in Medicine, Catholic University of the Sacred Heart, Largo F. Vito 1, 00168 Rome, Italy

**Keywords:** leadership effectiveness, healthcare settings, healthcare workers, private healthcare setting, public hospital, before–after, cross-sectional, leadership style

## Abstract

To work efficiently in healthcare organizations and optimize resources, team members should agree with their leader’s decisions critically. However, nowadays, little evidence is available in the literature. This systematic review and meta-analysis has assessed the effectiveness of leadership interventions in improving healthcare outcomes such as performance and guidelines adherence. Overall, the search strategies retrieved 3,155 records, and 21 of them were included in the meta-analysis. Two databases were used for manuscript research: PubMed and Scopus. On 16th December 2019 the researchers searched for articles published in the English language from 2015 to 2019. Considering the study designs, the pooled leadership effectiveness was 14.0% (95%CI 10.0–18.0%) in before–after studies, whereas the correlation coefficient between leadership interventions and healthcare outcomes was 0.22 (95%CI 0.15–0.28) in the cross-sectional studies. The multi-regression analysis in the cross-sectional studies showed a higher leadership effectiveness in South America (β = 0.56; 95%CI 0.13, 0.99), in private hospitals (β = 0.60; 95%CI 0.14, 1.06), and in medical specialty (β = 0.28; 95%CI 0.02, 0.54). These results encourage the improvement of leadership culture to increase performance and guideline adherence in healthcare settings. To reach this purpose, it would be useful to introduce a leadership curriculum following undergraduate medical courses.

## 1. Introduction

Over the last years, patients’ outcomes, population wellness and organizational standards have become the main purposes of any healthcare structure [1]. These standards can be achieved following evidence-based practice (EBP) for diseases prevention and care [2,3] and optimizing available economical and human resources [3,4], especially in low-industrialized geographical areas [5]. This objective could be reached with effective healthcare leadership [3,4], which could be considered a network whose team members followed leadership critically and motivated a leader’s decisions based on the organization’s requests and targets [6]. Healthcare workers raised their compliance towards daily activities in an effective leadership context, where the leader succeeded in improving membership and performance awareness among team members [7]. Furthermore, patients could improve their health conditions in a high-level leadership framework. [8] Despite the leadership benefits for healthcare systems’ performance and patients’ outcomes [1,7], professionals’ confidence would decline in a damaging leadership context for workers’ health conditions and performance [4,9,10]. On the other hand, the prevention of any detrimental factor which might worsen both team performance and healthcare systems’ outcomes could demand effective leadership [4,7,10]. However, shifting from the old and assumptive leadership into a more effective and dynamic one is still a challenge [4]. Nowadays, the available evidence on the impact and effectiveness of leadership interventions is sparse and not systematically reported in the literature [11,12].

Recently, the spreading of the Informal Opinion Leadership style into hospital environments is changing the traditional concept of leadership. This leadership style provides a leader without any official assignment, known as an “opinion leader”, whose educational and behavioral background is suitable for the working context. Its target is to apply the best practices in healthcare creating a more familiar and collaborative team [2]. However, Flodgren et al. reported that informal leadership interventions increased healthcare outcomes [2].

Nowadays, various leadership styles are recognized with different classifications but none of them are considered the gold standard for healthcare systems because of heterogenous leadership meanings in the literature [4,5,6,12,13]. Leadership style classification by Goleman considered leaders’ behavior [5,13], while Chen DS-S proposed a traditional leadership style classification (charismatic, servant, transactional and transformational) [6].

Even if leadership style improvement depends on the characteristics and mission of a workplace [6,13,14], a leader should have both a high education in healthcare leadership and the behavioral qualities necessary for establishing strong human relationships and achieving a healthcare system’s goals [7,15]. Theoretically, any practitioner could adapt their emotive capacities and educational/working experiences to healthcare contexts, political lines, economical and human resources [7]. Nowadays, no organization adopts a policy for leader selection in a specific healthcare setting [15]. Despite the availability of a self-assessment leadership skills questionnaire for aspirant leaders and a pattern for the selection of leaders by Dubinsky et al. [15], a standardized and universally accepted method to choose leaders for healthcare organizations is still argued over [5,15].

Leadership failure might be caused by the arduous application of leadership skills and adaptive characteristics among team members [5,6]. One of the reasons for this negative event could be the lack of a standardized leadership program for medical students [16,17]. Consequently, working experience in healthcare settings is the only way to apply a leadership style for many medical professionals [12,16,17].

Furthermore, the literature data on leadership effectiveness in healthcare organizations were slightly significant or discordant in results. Nevertheless, the knowledge of pooled leadership effectiveness should motivate healthcare workers to apply leadership strategies in healthcare systems [12]. This systematic review and meta-analysis assesses the pooled effectiveness of leadership interventions in improving healthcare workers’ and patients’ outcomes.

## 2. Materials and Methods

A systematic review and meta-analysis was conducted according to the Preferred Reporting Items for Systematic reviews and Meta-Analyses (PRISMA) Statement guidelines [18]. The protocol was registered on the PROSPERO database with code CRD42020198679 on 15 August 2020. Following these methodological standards, leadership interventions were evaluated as the pooled effectiveness and influential characteristic of healthcare settings, such as leadership style, workplace, settings and the study period.

### 2.1. Data Sources and Search Strategy

PubMed and Scopus were the two databases used for the research into the literature. On 16th December 2019, manuscripts in the English language published between 2015 and 2019 were searched by specific MeSH terms for each dataset. Those for PubMed were “leadership” OR “leadership” AND “clinical” AND “outcome” AND “public health” OR “public” AND “health” OR “public health” AND “humans”. Those for Scopus were “leadership” AND “clinical” AND “outcome” AND “public” AND “health”.

### 2.2. Study Selection and Data Extraction

In accordance with the PRISMA Statement, the following PICOS method was used for including articles [18]: the target population was all healthcare workers in any hospital or clinical setting (Population); the interventions were any leader’s recommendation to fulfil quality standards or performance indexes of a healthcare system (Intervention) [19]; to be included, the study should have a control group or reference at baseline as comparison (Control); and any effectiveness measure in terms of change in adherence to healthcare guidelines or performances (Outcome). In detail, any outcome implicated into healthcare workers’ capacity and characteristics in reaching a healthcare systems purposes following the highest standards was considered as performance [19]. Moreover, whatever clinical practices resulted after having respected the recommendations, procedures or statements settled previously was considered as guideline adherence [20]. The selected study design was an observational or experimental/quasi-experimental study design (trial, case control, cohort, cross-sectional, before-after study), excluding any systematic reviews, metanalyses, study protocol and guidelines (Studies).

The leaders’ interventions followed Chen’s leadership styles classification [6]. According to this, the charismatic leadership style can be defined also as an emotive leadership because of members’ strong feelings which guide the relationship with their leader. Its purpose is the improvement of workers’ motivation to reach predetermined organizational targets following a leader’s planning strategies and foresights. Servant leadership style is a sharing leadership style in whose members can increase their skills and competences through steady leader support, and they have a role in an organization’s goals. The transformational leadership style focuses on practical aspects such as new approaches for problem solving, new interventions to reach purposes, future planning and viewpoints sharing. Originality in a transformational leadership style has a key role of improving previous workers’ and healthcare system conditions in the achievement of objectives. The transactional leadership style requires a working context where technical skills are fundamental, and whose leader realizes a double-sense sharing process of knowledge and tasks with members. Furthermore, workers’ performances are improved through a rewarding system [6].

In this study, the supervisor trained the research team for practical manuscript selection and data extraction. The aim was to ensure data homogeneity and to check the authors’ procedures for selection and data collection. The screening phase was performed by four researchers reading each manuscript’s title and abstract independently and choosing to exclude any article that did not fulfill the inclusion criteria. Afterwards, the included manuscripts were searched for in the full text. They were retrieved freely, by institutional access or requesting them from the authors.

The assessment phase consisted of full-text reading to select articles following the inclusion criteria. The supervisor solved any contrasting view about article selection and variable selection.

The final database was built up by collecting the information from all included full-text articles: author, title, study year, year of publication, country/geographic location, study design, viability and type of evaluation scales for leadership competence, study period, type of intervention to improve leadership awareness, setting of leader intervention, selection modality of leaders, leadership style adopted, outcomes assessed such as guideline adherence or healthcare workers’ performance, benefits for patients’ health or patients’ outcomes improvement, public or private hospitals or healthcare units, ward specialty, intervention in single specialty or multi-professional settings, number of beds, number of healthcare workers involved in leadership interventions and sample size.

Each included article in this systematic review and meta-analysis received a standardized quality score for the specific study design, according to Newcastle–Ottawa, for the assessment of the quality of the cross-sectional study, and the Study Quality Assessment Tools by the National Heart, Lung, and Blood Institute were used for all other study designs [21,22].

### 2.3. Statistical Data Analysis

The manuscripts metadata were extracted in a Microsoft Excel spreadsheet to remove duplicate articles and collect data. The included article variables for the quantitative meta-analysis were: first author, publication year, continent of study, outcome, public or private organization, hospital or local healthcare unit, surgical or non-surgical ward, multi- or single-professionals, ward specialty, sample size, quality score of each manuscript, leadership style, year of study and study design.

The measurement of the outcomes of interest (either performance or guidelines adherence) depended on the study design of the included manuscripts in the meta-analysis:for cross-sectional studies, the outcome of interest was the correlation between leadership improvement and guideline adherence or healthcare performance;the outcome derived from before–after studies or the trial was the percentage of leadership improvement intervention in guideline adherence or healthcare performance;the incidence occurrence of improved results among exposed and not exposed healthcare workers of leadership interventions and the relative risks (RR) were the outcomes in cohort studies;the odds ratio (OR) between the case of healthcare workers who had received a leadership intervention and the control group for case-control studies.

Pooled estimates were calculated using both the fixed effects and DerSimonian and Laird random effects models, weighting individual study results by the inverse of their variances [23]. Forest plots assessed the pooled estimates and the corresponding 95%CI across the studies. The heterogeneity test was performed by a chi-square test at a significance level of *p* < 0.05, reporting the I^2^ statistic together with a 25%, 50% or 75% cut-off, indicating low, moderate, and high heterogeneity, respectively [24,25].

Subgroup analysis and meta-regression analyses explored the sources of significant heterogeneity. Subgroup analysis considered the leadership style (charismatic, servant, transactional and transformational), continent of study (North America, Europe, Oceania), median cut-off year of study conduction (studies conducted between 2005 and 2011 and studies conducted between 2012 and 2019), type of hospital organization (public or private hospital), type of specialty (surgical or medical specialty) and type of team (multi-professional or single-professional team).

Meta-regression analysis considered the following variables: year of starting study, continent of study conduction, public or private hospital, surgical or non-surgical specialty ward, type of healthcare service (hospital or local health unit), type of healthcare workers involved (multi- or single-professional), leadership style, and study quality score. All variables included in the model were relevant in the coefficient analysis.

To assess a potential publication bias, a graphical funnel plot reported the logarithm effect estimate and related the standard error from each study, and the Egger test was performed [26,27].

All data were analyzed using the statistical package STATA/SE 16.1 (StataCorp LP, College 482 Station, TX, USA), with the “metan” command used for meta-analysis, and “metafunnel”, “metabias” and “confunnel” for publication bias assessment [28].

## 3. Results

### 3.1. Studies Characteristics

Overall, the search strategies retrieved 3,155 relevant records. After removing 570 (18.1%) duplicates, 2,585 (81.9%) articles were suitable for the screening phase, of which only 284 (11.0%) articles were selected for the assessment phase. During the assessment phase, 263 (92.6%) articles were excluded. The most frequent reasons of exclusion were the absence of relevant outcomes (*n* = 134, 51.0%) and other study designs (*n* = 61, 23.2%). Very few articles were rejected due to them being written in another language (*n* = 1, 0.4%), due to the publication year being out of 2015–2019 (*n* = 1, 0.4%) or having an unavailable full text (*n* = 3, 1.1%).

A total of 21 (7.4%) articles were included in the qualitative and quantitative analysis, of which nine (42.9%) were cross-sectional studies and twelve (57.1%) were before and after studies (Figure 1).

The number of healthcare workers enrolled was 25,099 (median = 308, IQR = 89–1190), including at least 2,275 nurses (9.1%, median = 324, IQR = 199–458). Most of the studies involved a public hospital (*n* = 16, 76.2%). Among the studies from private healthcare settings, three (60.0%) were conducted in North America. Articles which analyzed servant and charismatic leadership styles were nine (42.9%) and eight (38.1%), respectively. Interventions with a transactional leadership style were examined in six (28.6%) studies, while those with a transformational leadership style were examined in five studies (23.8%). Overall, 82 healthcare outcomes were assessed and 71 (86.6%) of them were classified as performance. Adherence-to-guidelines outcomes were 11 (13.4%), which were related mainly to hospital stay (*n* = 7, 64.0%) and drug administration (*n* = 3, 27.0%). Clements et al. and Lornudd et al. showed the highest number of outcomes, which were 19 (23.2%) and 12 (14.6%), respectively [29,30].

### 3.2. Leadership Effectiveness in before–after Studies

Before–after studies (Appendix A) involved 22,241 (88.6%, median = 735, IQR = 68–1273) healthcare workers for a total of twelve articles, of which six (50.0%) consisted of performance and five (41.7%) of guidelines adherence and one (8.3%) of both outcomes. Among healthcare workers, there were 1,294 nurses (5.8%, median = 647, IQR = 40–1,254). Only the article by Savage et al. reported no number of involved healthcare workers [31].

The number of studies conducted after 2011 or between 2012–2019 was seven (58.3%), while only one (8.3%) article reported a study beginning both before and after 2011. Most of studies were conducted in Northern America (*n* = 5, 41.7%). The servant leadership style and charismatic leadership style were the most frequently implemented, as reported in five (41.7%) and four (33.3%) articles, respectively. Only one (8.3%) study adopted a transformational leadership style.

The pooled effectiveness of leadership was 14.0% (95%CI 10.0–18.0%), with a high level of heterogeneity (I^2^ = 99.9%, *p* < 0.0001) among the before–after studies (Figure 2).

The highest level of effectiveness was reported by Weech-Maldonado R et al. with an effectiveness of 199% (95%CI 183–215%) based on the Cultural Competency Assessment Tool for Hospitals (CCATH) [39]. The effectiveness of leadership changed in accordance with the leadership style (Appendix A) and publication bias (Appendix A).

Multi-regression analysis indicated a negative association between leadership effectiveness and studies from Oceania, but this result was not statistically significant (β = −0.33; 95% IC −1.25, 0.59). On the other hand, a charismatic leadership style affected healthcare outcomes positively even if it was not statistically relevant (β = 0.24; 95% IC −0.69, 1.17) (Table 1).

### 3.3. Leadership Effectiveness in Cross Sectional Studies

A total of 2858 (median = 199, IQR = 110–322) healthcare workers were involved in the cross-sectional studies (Appendix A), of which 981 (34.3%) were nurses. Most of the studies were conducted in Asia (*n* = 4, 44.4%) and North America (*n* = 3, 33.3%). All of the cross-sectional studies regarded only the healthcare professionals’ performance. Multi-professional teams were involved in seven (77.8%) studies, and they were more frequently conducted in both medical and surgical wards (*n* = 6, 66.7%). The leadership styles were equally distributed in the articles and two (22.2%) of them examined more than two leadership styles at the same time.

The pooled effectiveness of the leadership interventions in the cross-sectional studies had a correlation coefficient of 0.22 (95%CI 0.15–0.28), whose heterogeneity was remarkably high (I^2^ = 96.7%, *p* < 0.0001) (Figure 3).

The effectiveness of leadership in the cross-sectional studies changed in accordance with the leadership style (Appendix A) and publication bias (Appendix A).

Multi-regression analysis showed a higher leadership effectiveness in studies conducted in South America (β = 0.56 95%CI 0.13–0.99) in private hospitals (β = 0.60; 95%CI 0.14–1.06) and in the medical vs. surgical specialty (β = −0.22; 95%CI −0.54, −0.02) (Table 2).

## 4. Discussion

Leadership effectiveness in healthcare settings is a topic that is already treated in a quantitative matter, but only this systematic review and meta-analysis showed the pooled effectiveness of leadership intervention improving some healthcare outcomes such as performance and adherence to guidelines. However, the assessment of leadership effectiveness could be complicated because it depends on the study methodology and selected outcomes [12]. Health outcomes might benefit from leadership interventions, as Flodgren et al. was concerned about opinion leadership [2], whose adhesion to guidelines increased by 10.8% (95% CI: 3.5–14.6%). On the other hand, other outcomes did not improve after opinion leadership interventions [2]. Another review by Ford et al. about emergency wards reported a summary from the literature data which acknowledged an improvement in trauma care management through healthcare workers’ performance and adhesion to guidelines after effective leadership interventions [14]. Nevertheless, some variables such as collaboration among different healthcare professionals and patients’ healthcare needs might affect leadership intervention effectiveness [14]. Therefore, a defined leadership style might fail in a healthcare setting rather than in other settings [5,13,14].

The leadership effectiveness assessed through cross-sectional studies was higher in South America than in other continents. A possible explanation of this result could be the more frequent use of a transactional leadership style in this area, where the transactional leadership interventions were effective at optimizing economic resources and improving healthcare workers’ performance through cash rewards [48]. Financing methods for healthcare organizations might be different from one country to another, so the effectiveness of a leadership style can change. Reaching both economic targets and patients’ wellness could be considered a challenge for any leadership intervention [48], especially in poorer countries [5].

This meta-analysis showed a negative association between leadership effectiveness and studies by surgical wards. Other research has supported these results, which reported surgical ward performance worsened in any leadership context (charismatic, servant, transactional, transformational) [47]. In those workplaces, adopting a leadership style to improve surgical performance might be challenging because of nervous tension and little available time during surgical procedures [47]. On the other hand, a cross-sectional study declared that a surgical team’s performance in private surgical settings benefitted from charismatic leadership-style interventions [42]. This style of leadership intervention might be successful among a few healthcare workers [42], where creating relationships is easier [6]. Even a nursing team’s performance in trauma care increased after charismatic leadership-style interventions because of better communicative and supportive abilities than certain other professional categories [29,47]. However, nowadays there is no standardized leadership in healthcare basic courses [5,6,12]. Consequently, promoting leadership culture after undergraduate medical courses could achieve a proper increase in both leadership agreement and working wellness as well as a higher quality of care. [17]. Furthermore, for healthcare workers who have already worked in a healthcare setting, leadership improvement could consist of implementing basic knowledge on that topic. Consequently, they could reach a higher quality of care practice through working wellness [17] and overcoming the lack of previous leadership training [17].

Although very few studies have included in a meta-analysis examined in private healthcare settings [35,38,40,41,42], leadership interventions had more effectiveness in private hospitals than in public hospitals. This result could be related to the continent of origin, and indeed 60.0% of these studies were derived from North America [38,41,42], where patients’ outcomes and healthcare workers’ performance could influence available hospital budgets [38,40,41,42], especially in peripheral healthcare units [38,41]. Private hospitals paid more attention to the cost-effectiveness of any healthcare action and a positive balance of capital for healthcare settings might depend on the effectiveness of leadership interventions [40,41,42]. Furthermore, private healthcare assistance focused on nursing performance because of its impact on both a patients’ and an organizations’ outcomes. Therefore, healthcare systems’ quality could improve with effective leadership actions for a nursing team [40].

Other factors reported in the literature could affect leadership effectiveness, although they were not examined in this meta-analysis. For instance, professionals’ specialty and gender could have an effect on these results and shape leadership style choice and effectiveness [1]. Moreover, racial differences among members might influence healthcare system performance. Weech-Maldonado et al. found a higher compliance and self-improvement by black-race professionals than white ones after transactional leadership interventions [39].

Healthcare workers’ and patients’ outcomes depended on style of leadership interventions [1]. According to the results of this meta-analysis, interventions conducted by a transactional leadership style increased healthcare outcomes, though nevertheless their effectiveness was higher in the cross-sectional studies than in the before–after studies. Conversely, the improvement by a transformational leadership style was higher in before–after studies than in the cross-sectional studies. Both a charismatic and servant leadership style increased effectiveness more in the cross-sectional studies than in the before–after studies. This data shows that any setting required a specific leadership style for improving performance and guideline adherence by each team member who could understand the importance of their role and their tasks [1]. Some outcomes had a better improvement than others. Focusing on Savage et al.’s outcomes, a transformational leadership style improved checklist adherence [31]. The time of patients’ transport by Murphy et al. was reduced after conducting interventions based on a charismatic leadership style [37]. Jodar et al. showed that performances were elevated in units whose healthcare workers were subjected to transactional and transformational leadership-style interventions [1].

These meta-analysis results were slightly relevant because of the high heterogeneity among the studies, as confirmed by both funnel plots. This publication bias might be caused by unpublished articles due to either lacking data on leadership effectiveness, failing appropriate leadership strategies in the wrong settings or non-cooperating teams [12]. The association between leadership interventions and healthcare outcomes was slightly explored or gave no statistically significant results [12], although professionals’ performance and patients’ outcomes were closely related to the adopted leadership style, as reported by the latest literature sources [7]. Other aspects than effectiveness should be investigated for leadership. For example, the evaluation of the psychological effect of leadership should be explored using other databases.

The study design choice could affect the results about leadership effectiveness, making their detection and their statistical relevance tough [12]. Despite the strongest evidence of this study design [50], nowadays, trials about leadership effectiveness on healthcare outcomes are lacking and have to be improved [12]. Notwithstanding, this analysis gave the first results of leadership effectiveness from the available study designs.

Performance and adherence to guidelines were the main two outcomes examined in this meta-analysis because of their highest impact on patients, healthcare workers and hospital organizations. They included several other types of outcomes which were independent each other and gave different effectiveness results [12]. The lack of neither an official classification nor standardized guidelines explained the heterogeneity of these outcomes. To reach consistent results, they were classified into performance and guideline adherence by the description of each outcome in the related manuscripts [5,6,12].

Another important aspect is outcome assessment after leadership interventions, which might be fulfilled by several standardized indexes and other evaluation methods [40,41]. Therefore, leadership interventions should be investigated in further studies [5], converging on a univocal and official leadership definition and classification to obtain comparable results among countries [5,6,12].

## 5. Conclusions

This meta-analysis gave the first pooled data estimating leadership effectiveness in healthcare settings. However, some of them, e.g., surgery, required a dedicated approach to select the most worthwhile leadership style for refining healthcare worker performances and guideline adhesion. This can be implemented using a standardized leadership program for surgical settings.

Only cross-sectional studies gave significant results in leadership effectiveness. For this reason, leadership effectiveness needs to be supported and strengthened by other study designs, especially those with the highest evidence levels, such as trials. Finally, further research should be carried out to define guidelines on leadership style choice and establish shared healthcare policies worldwide.

## Figures and Tables

**Figure 1 ijerph-19-10995-f001:**
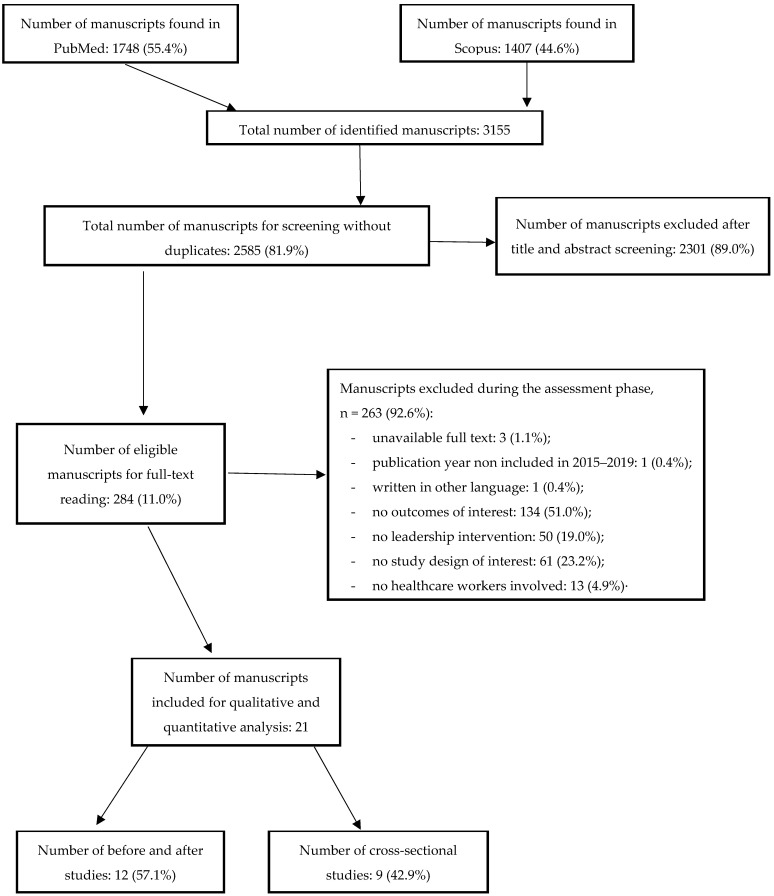
Flow-chart of selection manuscript phases for systematic review and meta-analysis on leadership effectiveness in healthcare workers.

**Figure 2 ijerph-19-10995-f002:**
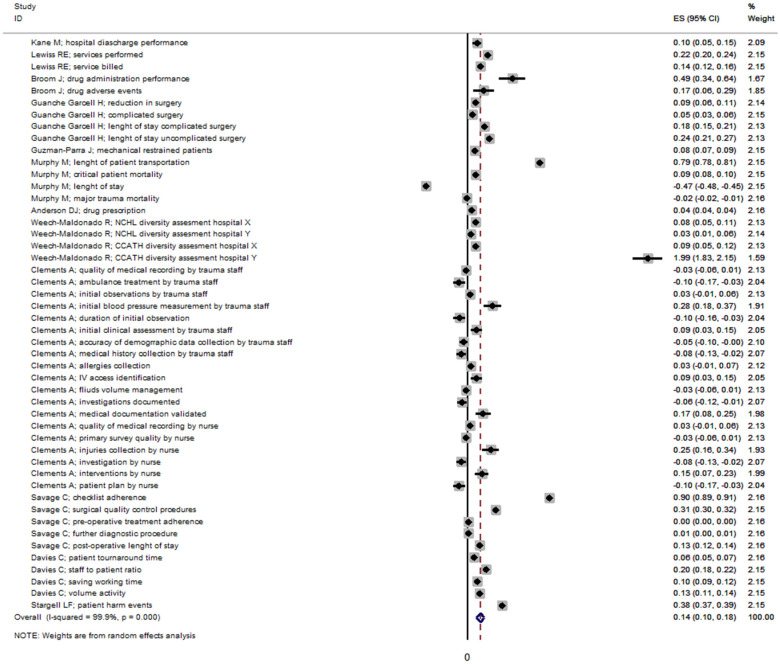
Effectiveness of leadership in before after studies. Dashed line represents the pooled effectiveness value [29,31,32,33,34,35,36,37,38,39,40,41].

**Figure 3 ijerph-19-10995-f003:**
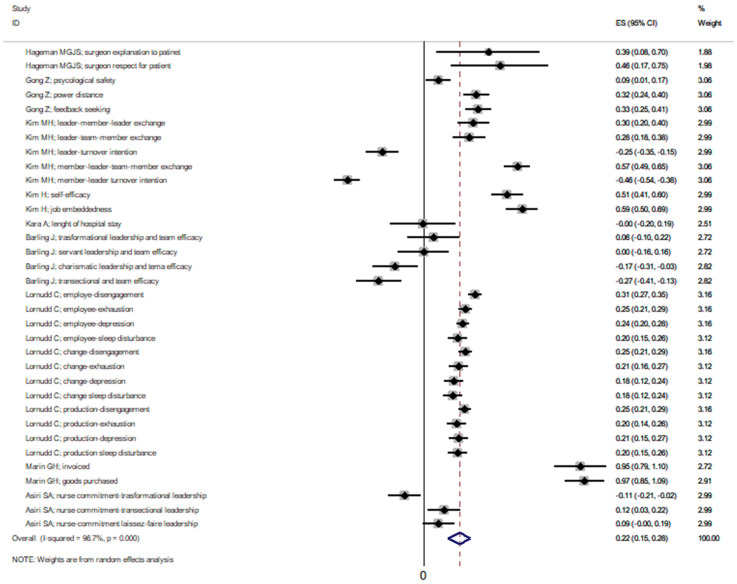
Effectiveness of leadership in cross-sectional studies. Dashed line represents the pooled effectiveness value [30,42,43,44,45,46,47,48,49].

**Table 1 ijerph-19-10995-t001:** Correlation coefficients and multi-regression analysis of leadership effectiveness in before–after studies.

Variables	Correlation Coefficient	Beta Coefficient	95% CI	
Studies conducted between 2012–2019 vs. 2005–2011 years	−0.26	−0.09	−0.42	0.24
North American continent vs. others	0.27	−0.04	−0.82	0.75
Oceanian continent vs. others	−0.26	−0.33	−1.25	0.59
European continent vs. others	0.07	−0.27	−1.12	0.58
Public hospital vs. private hospital	0.01			
Surgical specialty vs. non-surgical specialty	−0.21	−0.05	−0.85	0.75
Leadership style transformational vs. other styles	0.12	0.32	−0.47	1.11
Leadership style charismatic vs. other styles	−0.23	0.24	−0.69	1.17
Leadership style transactional vs. other styles	0.25	0.25	−0.40	0.91

**Table 2 ijerph-19-10995-t002:** Multi-regression analysis of leadership effectiveness in cross-sectional studies.

Variables	Correlation Coefficient	Beta Coefficient	95% CI	
Studies conducted between 2012–2019 vs. 2005–2011 years	−0.31	−0.09	−0.27	0.10
South American continent vs. others	0.63	0.56 *	0.13	0.99
Private hospital vs. public hospital	0.17	0.60 *	0.14	1.06
Surgical specialty vs. non-surgical specialty	−0.22	−0.28 *	−0.54	−0.02
Leadership style transformational vs. other styles	0.41	0.16	−0.14	0.46
Leadership style charismatic vs. other styles	−0.14	−0.04	−0.26	0.18
Leadership style transactional vs. other styles	−0.11	0.01	−0.21	0.23
Multiprofessional team vs. single professional team	0.04			

* 0.05 ≤ *p* < 0.01.

## Data Availability

Data will be available after writing correspondence to the author.

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
