# Peer review of "Leadership Effectiveness in Healthcare Settings: A Systematic Review and Meta-Analysis of Cross-Sectional and Before–After Studies"

_ijerph, 2022, doi:10.3390/ijerph191710995_

Round 1
Reviewer 1 Report (Previous Reviewer 2)
Authors considerably improved their manuscript, especially the method and discussion parts. I only have some minor comments.
A descriptive statistics table would help understand the characteristics of the samples.
Figure 1 needs to improve the quality of the figure.
Authors should clarify what surgical and medical specialties refer to as they are meaningful (significant) variables in the analysis. Readers may need some examples at least. The term "medical" sounds too broad to me. I recommend using "surgical and non-surgical" instead of "surgical and medical," but it's the authors' call.
Author Response
Authors considerably improved their manuscript, especially the method and discussion parts. I only have some minor comments.
A descriptive statistics table would help understand the characteristics of the samples.
I agree with your request, indeed the descriptive analysis of all the included articles was already included in the supplementary fiie (see supllementary Table 1).
Figure 1 needs to improve the quality of the figure.
I recognize the need to improve the figure’s quality. Thank you so much for your notification.
Authors should clarify what surgical and medical specialties refer to as they are meaningful (significant) variables in the analysis. Readers may need some examples at least. The term "medical" sounds too broad to me. I recommend using "surgical and non-surgical" instead of "surgical and medical," but it's the authors' call.
I changed the term medical in non surgical as you suggested.
Reviewer 2 Report (New Reviewer)
The topic of the manuscript is very important and current, but many methodological issues need to be addressed more deeply and a lot of clarifications and justifications are needed, such as defining the main concepts, research question, updating searches (done in 2019), defining interventions and outcomes clearly and inclusion and exclusion criteria e.g. by using PICOs format, carrying out the review according to some clear guidelines, clarifying the search strategy and describing the quality assessment in more detail. In addition, the manuscript seems unfinished because it has comments and side notes, and it seems that the manuscript is not finished. Please finish the manuscript carefully.
Author Response
The topic of the manuscript is very important and current, but many methodological issues need to be addressed more deeply and a lot of clarifications and justifications are needed, such as defining the main concepts, research question, updating searches (done in 2019), defining interventions and outcomes clearly and inclusion and exclusion criteria e.g. by using PICOs format, carrying out the review according to some clear guidelines, clarifying the search strategy and describing the quality assessment in more detail. In addition, the manuscript seems unfinished because it has comments and side notes, and it seems that the manuscript is not finished. Please finish the manuscript carefully.
Thank you for these suggestions, to whom I will try to answer. First of all, as far as updating searches (done in 2019) concerned, the research was not updated after 2019 due to COVID-19 pandemic. Indeed the pandemic had changed many aspects in healthcare systems management and leadership.
Main concepts
The main concepts of this work are that outcomes such as healthcare workers’ performance and adherence to guidelines improve in settings where leadership interventions had been applied, as reported into results section “The pooled effectiveness of leadership was 14.0% (95%CI 10.0-18.0%), with a high level of heterogeneity (I2=99.9%, p<0.0001) among before after studies”, “The pooled effectiveness of leadership interventions in cross sectional studies had a correlation coefficient of 0.22 (95%CI 0.15-0.28), […]”. Moreover, the result of leadership effectiveness changes considering other characteristics, such as the setting (surgical and non-surgical), continents, and leadership styles.
Research question
The research question is in the final part of the introduction: “This systematic review and meta-analysis assesses the pooled effectiveness of leadership interventions in improving healthcare workers’ and patients’ outcomes.” which is linked to the first part of the same section “Over the last years, patients' outcomes, population wellness and organization standards are becoming the main purposes of any healthcare structure [1], which have to be achieved following evidence-based practice (EBP) for diseases prevention and care [2,3] and optimizing available economical and human [3,4], specially in low industrialized geographical areas [5].”
Defining interventions and outcomes clearly and inclusion and exclusion criteria e.g. by using PICOs format
You can find the explanation of interventions and outcomes into materials and methods section: “In accordance with the PRISMA Statement, the following PICOS method was used for including articles [18]: the target population was all healthcare workers in any hospital or clinical setting (Population); the interventions were any leaders’ recommendation to fulfil quality standards or performance indexes of the healthcare systems (Intervention) [19]; to be included, the study should have a control group or reference at baseline as comparison (Control); any effectiveness measure in terms of change in adherence to healthcare guide-lines or performances (Outcome). In detail, any outcome implicated into healthcare work-ers' capacity and characteristics in reaching healthcare systems purposes following the highest standards was considered as performance [19]. Moreover, whatever clinical prac-tices results after having respected recommendations, procedures, or statements settled previously was considered as guidelines adherence [20]. Selected study design was obser-vational or experimental/quasi-experimental study design (trial, case control, cohort, cross-sectional, before-after study), excluding any systematic reviews, metanalyses, study protocol and guidelines (Studies).”
Carrying out the review according to some clear guidelines
This study was carried out according to the PRISMA Statement guidelines, which is the guideline to perform this study design, as it was written in the first part of material and methods: “A systematic review and meta-analysis was conducted according to the Preferred Reporting Items for Systematic reviews and Meta-Analyses (PRISMA) Statement”
Clarifying the search strategy
The search strategy was using MeSH terms in the most common online database PubMed and Scopus, as it was described in the following part of materials and methods: “PubMed and Scopus were the two databases used for research in the literature. On 16th December 2019, manuscripts in English language published between 2015 and 2019 were searched by specific MeSH terms for each dataset: those for PubMed were "leadership" OR "leadership" AND “clinical” AND “outcome” AND "public health" OR "public" AND "health" OR "public health" AND "humans", instead those for Scopus were “leadership” AND “clinical” AND “outcome” AND “public” AND “health”.
Describing the quality assessment in more detail
Each article received the quality score according to the references number 21 and 22 as reported in the methods section. “Each included article in this systematic review and meta-analysis received a standardized quality score for specific study design, according to Newcastle-Ottawa for the assessment of cross-sectional studies quality and Study Quality Assessment Tools by National Heart, Lung, and Blood Institute for all other study designs [21,22].”
Reviewer 3 Report (New Reviewer)
1. Originality: Does the paper contain new and significant information adequate to justify publication?: Yes. he authors identified an interesting topic and provided significant information on the relation of “The leadership effectiveness in healthcare settings: a systematic review and meta-analysis of cross-sectional and before-after studies. However, there are some issues that need clarification. More importantly, the discussion regarding the research gaps is missing!
2. Relationship to Literature: Does the paper demonstrate an adequate understanding of the relevant literature in the field and cite an appropriate range of literature sources? Is any significant work ignored?: The literature review section is fine. Add the below mentioned article regarding leadership styles.
3. Methodology: Is the paper's argument built on an appropriate base of theory, concepts, or other ideas? Has the research or equivalent intellectual work on which the paper is based been well designed? Are the methods employed appropriate?: The paper is well structured and follows the standards of presenting hypothesis and research design.
4. Results: Are results presented clearly and analysed appropriately? Do the conclusions adequately tie together the other elements of the paper?: The results section is fine.
5. Implications for research, practice and/or society: Does the paper identify clearly any implications for research, practice and/or society? Does the paper bridge the gap between theory and practice? How can the research be used in practice (economic and commercial impact), in teaching, to influence public policy, in research (contributing to the body of knowledge)? What is the impact upon society (influencing public attitudes, affecting quality of life)? Are these implications consistent with the findings and conclusions of the paper?: The implications of the research is not clear. Kindly improve this part.
6. Quality of Communication: Does the paper clearly express its case, measured against the technical language of the field and the expected knowledge of the journal's readership? Has attention been paid to the clarity of expression and readability, such as sentence structure, jargon use, acronyms, etc.: A professional review of the language is strongly suggested because several parts of the text are unclear.
Author Response
- Originality: Does the paper contain new and significant information adequate to justify publication?: Yes. he authors identified an interesting topic and provided significant information on the relation of “The leadership effectiveness in healthcare settings: a systematic review and meta-analysis of cross-sectional and before-after studies. However, there are some issues that need clarification. More importantly, the discussion regarding the research gaps is missing!
Thank you for your observations. The research gap was implemented in the discussion section as following: “Another important aspect is outcomes assessment after leadership interventions, which might be fulfilled by several standardized indexes and other evaluation methods [37,39]. Therefore, leadership interventions should be investigated in further studies [5], converging on univocal and official leadership definition and classification to obtain com-parable results among countries [5,6,12].”
- Relationship to Literature: Does the paper demonstrate an adequate understanding of the relevant literature in the field and cite an appropriate range of literature sources? Is any significant work ignored?: The literature review section is fine. Add the below mentioned article regarding leadership styles.
The article regarding leadership styles is the following: Chen, D.S.-S. Leadership Styles and Organization Structural Configurations. 2006, 8. Other articles are quoted in the references section.
- Methodology: Is the paper's argument built on an appropriate base of theory, concepts, or other ideas? Has the research or equivalent intellectual work on which the paper is based been well designed? Are the methods employed appropriate?: The paper is well structured and follows the standards of presenting hypothesis and research design.
Thank you so much for appreciating our work.
- Results: Are results presented clearly and analysed appropriately? Do the conclusions adequately tie together the other elements of the paper?: The results section is fine.
Thank you so much.
- Implications for research, practice and/or society: Does the paper identify clearly any implications for research, practice and/or society? Does the paper bridge the gap between theory and practice? How can the research be used in practice (economic and commercial impact), in teaching, to influence public policy, in research (contributing to the body of knowledge)? What is the impact upon society (influencing public attitudes, affecting quality of life)? Are these implications consistent with the findings and conclusions of the paper?: The implications of the research is not clear. Kindly improve this part.
Thank you so much for your advice. We add some parts in the discussion section: “Furthermore, for healthcare workers who have already worked in healthcare setting, the leadership improvement could consist in implementing basic knowledge on that topic. Consequently, they would reach higher quality of care practice through working wellness [17] and overcoming the lack of previous leadership training [17].”
- Quality of Communication: Does the paper clearly express its case, measured against the technical language of the field and the expected knowled6ge of the journal's readership? Has attention been paid to the clarity of expression and readability, such as sentence structure, jargon use, acronyms, etc.: A professional review of the language is strongly suggested because several parts of the text are unclear.
We review all paper in order to improve the language.
Round 2
Reviewer 2 Report (New Reviewer)
Great work, I have no further comments. All the best for this manuscript.
This manuscript is a resubmission of an earlier submission. The following is a list of the peer review reports and author responses from that submission.
Round 1
Reviewer 1 Report
Thank you very much for reviewing this paper. I find the topic very interesting and the way of developing it new. As an expert in bibliometric analysis, I have not read many papers with this approach, so it seems interesting to me. It would be advisable for them to explain why they have applied this methodology versus other literature analysis techniques such as co-citation analysis (citespace) or keyword analysis (scimat). Finally, I recommend that the conclusions be further developed indicating what the future is in this field.
Author Response
Question (Q). Thank you very much for reviewing this paper. I find the topic very interesting and the way of developing it new. As an expert in bibliometric analysis, I have not read many papers with this approach, so it seems interesting to me. It would be advisable for them to explain why they have applied this methodology versus other literature analysis techniques such as co-citation analysis (citespace) or keyword analysis (scimat). Finally, I recommend that the conclusions be further developed indicating what the future is in this field.
Answer (A). I was pleased to read your remark. The methodology used to this systematic review and meta-analysis followed the instructions according to the PRISMA Statement, a sort of official guidelines to do this type of study design. Unlike bibliometric analysis, meta-analysis has as objective a single result by the pooled analysis of results retrieved from manuscripts which have common variables. This gives stronger evidence of a single outcome by several studies to develop guidelines or healthcare programs.
As far as the future of leadership, I changed the conclusion to fulfill your suggestion: “This meta-analysis gave the first pooled data to estimate leadership effectiveness. Some healthcare settings as surgery required a dedicated approach to adopt the most worthwhile leadership style and refine on healthcare workers performances and guidelines adhesion. This can be implemented with a standardized leadership program for surgical setting, in order to learn the positive benefits in the healthcare.
Furthermore, in this meta-analysis cross-sectional studies give significant results in assessing leadership effectiveness. To strengthen the evidence, leadership interventions should be further investigated with other study designs, producing higher levels evidences, such as trial. The further research could lead in the future to evidences use-ful to define guidelines on leadership adoption and developing healthcare policies.”
Reviewer 2 Report
I think this article needs further improvement in the following aspects:
The years they examined (2015-2019) are outdated, and the authors lack logical explanations why they include only those five years.
The abstract section needs to be considerably revised because it is hard to understand the main findings and conclusions by reading it. That may be because the results and implications of the article are not fully developed. What do the authors want readers to learn from this meta-analysis?
The authors did not sufficiently demonstrate how leadership effectiveness is defined/classified in the literature and the manuscript. Please elaborate.
There is no reasonable logic on why the authors divided the literature into before-after studies and cross-sectional studies. That does not offer much implication to readers. Why is the “before and after 2011” criterion important enough to be addressed in the manuscript? Why are regions significant enough to be discussed in the manuscript? There might be more meaningful ways for the authors to categorize the articles included in this meta-analysis.
To make the research results more credible, the authors should present the detailed findings (outcomes) of each study included in the meta-analysis may be helpful for readers.
The discussion part of the article needs to be further strengthened as there is little connection between the findings and conclusions, which simply state that undergraduate education may improve leadership culture. How would the results justify “the necessity of improving leadership culture starting from undergraduate medical courses for both increase quality of care and performance through working wellness”? (p.1)
Minor errors in general:
· What does “feelings making” (p.6) mean?
· The following sentence doesn’t sound professional and scholarly. “However, demonstrating leadership effectiveness is awkward.” (p.6)
· What does “conduction” mean in tables 1 & 2? There are many other typos in the tables, including those in supplementary (e.g., Sud America). Please check.
· Supplementary Figure 1 (article selection procedure) should be presented in the main manuscript.
· In Table 2, delete “Conduction after 2011 vs before 2011.”
· In Tables 1&2, indicate the statistical significance levels represented by *, **, ***.
Minor errors in supplementary Tables 1 &2:
· In Supplementary Table 1, separate the type of outcome from the first author. I also recommend rearranging the table by categorizing the types of outcomes.
· In Supplementary Table 2, include outcome type. Avoid using acronyms (e.g., Lhu)
· More detailed explanations of what quality score means are needed. For example, what does a quality score of 10 mean?
Author Response
I think this article needs further improvement in the following aspects:
- The years they examined (2015-2019) are outdated, and the authors lack logical explanations why they include only those five years.
- This meta-analysis started at the end of 2019. Because of COVID-19 pandemic, the inclusion of the years after 2019 could falsify the pooled results. For this reason, the examined period remained between 2015 and 2019.
- The abstract section needs to be considerably revised because it is hard to understand the main findings and conclusions by reading it. That may be because the results and implications of the article are not fully developed. What do the authors want readers to learn from this meta-analysis? Answer
- I found your request much appropriate for improving the meaning of the abstract. I rewrite as the following [row 22-41]: “To work efficiently in healthcare organizations and optimize resources, team members should agree with leader’s decisions critically. However, nowadays little evidence is available in the literature. This systematic review and meta-analysis has assessed leadership interventions effectiveness in improving healthcare outcomes such as performance and guidelines adherence. Overall, the search strategies retrieved 3,155 records and 21 of them were included in the meta-analysis. Two databases were used for manuscripts research: PubMed and Scopus. On 16th December 2019 the researchers searched for articles published in English language from 2015 to 2019. Considering study designs, the pooled leadership effectiveness was 14.0% (95%CI 10.0-18.0%) in before-after studies, whereas the correlation coefficient between leadership interventions and healthcare outcomes was 0.22 (95%CI 0.15-0.28) in cross sectional studies. The multi-regression analysis in cross-sectional studies showed a higher leadership effectiveness in South America (β=0.54; 95%CI 0.12, 0.95), in the private hospitals (β=0.87; 95%CI 0.31, 1.43), and in medical specialty (β=0.34; 95%CI 0.08, 0.60). These results encourage the improvement of leadership culture to increase performance and guidelines adherence in healthcare settings. To reach this purpose, it should be useful to introduce leadership curriculum since undergraduate medical courses”
- The authors did not sufficiently demonstrate how leadership effectiveness is defined/classified in the literature and the manuscript. Please elaborate.
- I understood your perplexity about the lack of definition or classification on leadership effectiveness in the literature. To improve the understanding of this, I introduce the following text in the manuscript [row 131-136]: “In detail, any outcome implicated into healthcare workers' capacity and characteristics to reach the purposes of the healthcare systems following the highest standards is considered as performance [19]. Whereas whatever clinical practices result after having respected recommendations, procedures, or statements settled previously was considered as guidelines adherence [20]”. Furthermore we specified better the outcomes’ results of interest [rows 189-201]: “The measures of the outcomes of interest (either performance or guidelines adherence) depended on the study design of the included manuscripts in the meta-analysis:
- for cross-sectional studies, the outcome of interest was the correlation be-tween leadership improvement and guidelines adherence or healthcare performance;
- the outcome derived from before after studies or trial was the percentage of leadership improvement intervention in guidelines adherence or healthcare performance;
- the incidence of improved results among exposed and not exposed healthcare workers of leadership interventions and the relative risks (RR) were the outcome in cohort studies;
- the odds ratio (OR) between case of healthcare workers who had received a leadership intervention and the control for case-control studies.”
- There is no reasonable logic on why the authors divided the literature into before-afterstudies and cross-sectionalstudies. That does not offer much implication to readers. Why is the “before and after 2011” criterion important enough to be addressed in the manuscript? Why are regions significant enough to be discussed in the manuscript? There might be more meaningful ways for the authors to categorize the articles included in this meta-analysis.
- All the manuscripts are classified into different study designs because each of them had a specific methodology to conduct the research. According to this, after reading materials and methods and results of each scientific paper, it was recognized the appropriate study design, which was attributed a quality score following The Newcastle–Ottawa Scale (NOS) for Assessing the Quality of Non-Randomized Studies in Meta-Analysis (number 19 of the reference list) and Quality Assessment Tool for Studies with No Control Group. Study Quality Assessment Tools (number 20 of the reference list). The studies designs which met the inclusion criteria was cross-sectional and before after studies and this is the reason way only them were included in the analysis. Furthermore they was analyzed separately due to different methodology used in each design of study.
As far as it concerned before and after 2011, the median year of study period was 2011. For this reason, the studies were classified into before and after 2011. This subdivision is really useful to evaluate the evolution of the leadership interventions effectiveness during an equal period of time for study coontribution.
Since regions or continents had different healthcare systems, health determinants, and population characteristics, we chose to use the variables subdivided by regions. Furthermore, the division of studies by region is useful to explain how to apply leadership interventions and to improve the outcomes.
Finally, the variables retrieved from all the included articles was planned according to the literature revision and following the PICOS described in the PRISMA Statement. Moreover, the plan of variable extraction was higher in number than variable included in the meta-regression analysis. This is due to the reason that not all studies collected all the planned variables.
- To make the research results more credible, the authors should present the detailed findings (outcomes) of each study included in the meta-analysis may be helpful for readers.
- Thank you for your observation. All the outcomes were summarized into performance and guidelines adherence, which are the main outcomes involved in patients’ health conditions and healthcare settings targets. To improve the understanding of this, I introduce the following text in the manuscript [row 131-136]: “In detail, any outcome implicated into healthcare workers' capacity and characteristics to reach the purposes of the healthcare systems following the highest standards is considered as performance [19]. Whereas whatever clinical practices result after having respected recommendations, procedures, or statements settled previously was considered as guidelines adherence [20]”.
The single outcome is deducted from the included manuscripts into meta-analysis: performance is the quality of daily tasks or results reached by healthcare workers; guidelines adherence is the capacity to follow several recommendations correctly. The classification of outcome by studies is already reported in supplementary table 1 and in supplementary table 2.
- The discussion part of the article needs to be further strengthened as there is little connection between the findings and conclusions, which simply state that undergraduate education may improve leadership culture. How would the results justify “the necessity of improving leadership culture starting from undergraduate medical courses for both increase quality of care and performance through working wellness”? (p.1)
- The discussion was implemented in order to highlight the importance of leadership knowledge implementation in the medical setting as following “However, nowadays there is no standardized leadership in the healthcare basic course [5,6,12]. Consequently, promoting leadership culture since undergraduate medical courses to achieve properly increase in both leadership agreement and working wellness, as well as reaching a higher quality of care. [17].” Furthermore, to have more connection between the discussion and the conclusion we rewrite the discussion s following: “This meta-analysis gave the first pooled data to estimate leadership effectiveness. Some healthcare settings as surgery required a dedicated approach to adopt the most worthwhile leadership style and refine on healthcare workers performances and guidelines adhesion. This can be implemented with a standardized leadership program for surgical setting, in order to learn the positive benefits in the healthcare.
Furthermore, in this meta-analysis cross-sectional studies give significant results in assessing leadership effectiveness. To strengthen the evidence, leadership interventions should be further investigated with other study designs, producing higher levels evidences, such as trial. The further research could lead in the future to evidences use-ful to define guidelines on leadership adoption and developing healthcare policies”.
Minor errors in general:
- What does “feelings making” (p.6) mean?
- We changed this expression in order to make more readable as “creating relationships” [row 293].
- The following sentence doesn’t sound professional and scholarly. “However, demonstrating leadership effectiveness is awkward.” (p.6)
- Thanking for this, I changed it into “However, the assessment of leadership effectiveness could be complicated because it depends by the study methodology and the outcomes selected [12].”[rows 264-266].
- What does “conduction” mean in tables 1 & 2? There are many other typos in the tables, including those in supplementary (e.g., Sud America). Please check.
- Thank you for your observation. I changed some of them:
- “conduction” into “study period” in both Table 1 and 2;
- “Continent Nord American vs other continents” into “North American continent vs others” in Table 1;
- “Continent Oceania vs other continents” into “Oceanian continent vs others” in Table 1;
- “Continent Europe vs other continents” into “European continent vs others” in Table 1;
- “Continent Sud American vs other continents” into “South American continent vs others” in Table 2;
- “Sud America” into “South America” in Supplementary Table 2;
- Supplementary Figure 1 (article selection procedure) should be presented in the main manuscript.
- A. We fully agree with you and added Figure 1 into the manuscript with an adequate description.
- In Table 2, delete “Conduction after 2011 vs before 2011.”
- For us, conduction was the study period, so it cannot cancel. Nevertheless, I prefer substituting this with study period.
- In Tables 1&2, indicate the statistical significance levels represented by *, **, ***.
- The significance level was added in footnotes of table 2 according to the following rule: * 0.05≤p<0.01; **0.01≤p<0.001; ***p≤0.001
- Minor errors in supplementary Tables 1 &2:
- In Supplementary Table 1, separate the type of outcome from the first author. I also recommend rearranging the table by categorizing the types of outcomes.
- Your suggestion for improving the meaning of Supplementary Table 1 is convenient, dividing adhesion to guidelines from performance outcome. On the contrary, Table 2 contains only performance outcomes, so as it was described into the title of Table 2.
- In Supplementary Table 2, include outcome type. Avoid using acronyms (e.g., Lhu)
- I agree with you for this correction, so I write the full name of local health unit, Furthermore Table 2 contains only performance outcomes, so as it was described into the title of Table 2.
- More detailed explanations of what quality score means are needed. For example, what does a quality score of 10 mean?
- I am pleased to receive such observation from you. The quality score for each study design follows The Newcastle–Ottawa Scale (NOS) for Assessing the Quality of Non-Randomized Studies in Meta-Analysis (number 19 of the reference list) and Quality Assessment Tool for Studies with No Control Group. Study Quality Assessment Tools (number 20 of the reference list). Considering each study design, the quality score ranged from 0 (for all of them) to 9 for case-control and cohort study, 10 for cross-sectional study, 12 for before after study, and 14 for trial study”. The quality score is useful in identifying the methodological level according to which were designed and realized studies. The level of accuracy could have an influence on results obtained and for this reason the quality score for each study was evaluated and used in the meta-regression analysis.